# Fine-Tuning of Atomic Energies in Relativistic Multiconfiguration Calculations

**Yanting Li [1,2]**, **Gediminas Gaigalas [3]**, **Wenxian Li [4]**, **Chongyang Chen [1]** and **Per Jönsson [2,*]**

1   Shanghai EBIT Lab, Key Laboratory of Nuclear Physics and Ion-Beam Application, Institute of Modern Physics, Department of Nuclear Science and Technology, Fudan University, Shanghai 200433, China; 18110200005@fudan.edu.cn (Y.L.); chychen@fudan.edu.cn (C.C.)
2   Department of Materials Science and Applied Mathematics, Malmö University, 20506 Malmö, Sweden
3   Institute of Theoretical Physics and Astronomy, Vilnius University, 010222 Vilnius, Lithuania; gediminas.gaigalas@tfai.vu.lt
4   National Astronomical Observatories, Chinese Academy of Sciences, Beijing 100012, China; wxli@nao.cas.cn
*   Correspondence: per.jonsson@mau.se

**Abstract:** Ab initio calculations sometimes do not reproduce the experimentally observed energy separations at a high enough accuracy. Fine-tuning of diagonal elements of the Hamiltonian matrix is a process which seeks to ensure that calculated energy separations of the states that mix are in agreement with experiment. The process gives more accurate measures of the mixing than can be obtained in ab initio calculations. Fine-tuning requires the Hamiltonian matrix to be diagonally dominant, which is generally not the case for calculations based on $jj$-coupled configuration state functions. We show that this problem can be circumvented by a method that transforms the Hamiltonian in $jj$-coupling to a Hamiltonian in $LSJ$-coupling for which fine-tuning applies. The fine-tuned matrix is then transformed back to a Hamiltonian in $jj$-coupling. The implementation of the method into the General Relativistic Atomic Structure Package is described and test runs to validate the program operations are reported. The new method is applied to the computation of the $2s^2\,^1S_0 - 2s2p\,^{1,3}P_1$ transitions in C III and to the computation of Rydberg transitions in B I, for which the $2s2p^2\,^2S_{1/2}$ perturber enters the $2s^2ns\,^2S_{1/2}$ series. Improved convergence patterns and results are found compared with ab initio calculations.

**Keywords:** fine-tuning; multiconfiguration Dirac–Hartree–Fock; $jj$-coupling; $LSJ$-coupling; coupling transformation

## 1. Introduction

Fine-tuning of diagonal elements of the Hamiltonian matrix is a process which seeks to ensure that calculated energy separations of the states that mix are in agreement with experiment. The process gives more accurate measures of the mixing than can be obtained in an ab initio calculation and leads to more stable convergence patterns of calculated transition parameters as the calculation is systematically improved. Starting from already extensive calculations, fine-tuning has been used very successfully by Hibbert and co-workers to compute highly accurate transition parameters for both intercombination and allowed transitions [1–4]. Fine-tuning can, among other things, be used to compensate for the fact that energies of different $LS$-terms, depending on if the spins of the outer electrons are aligned or not, converge at different rates with respect to the increasing active set of orbitals building the wave functions, see for example ([5], pp. 96–97). It thus gives more accurate term mixings [6]. Fine-tuning can also be used to correctly position perturbers within a Rydberg series, something that can have very large effects on the transition rates, as well as other properties such as fine- and hyperfine structures, for neighboring states [7,8].

Fine-tuning has been applied to $LSJ$-coupled multiconfiguration Hartree–Fock (MCHF) and configuration interaction (CI) calculations within the Breit–Pauli approximation, and it

is available as an option in the frequently used computer codes CIV3 and ATSP2K [9,10]. The tuning process has, so far, not been used for *jj*-coupled multiconfiguration Dirac–Hartree–Fock (MCDHF) or relativistic configuration interaction (RCI) calculations, the reason being that the Hamiltonian matrix is not diagonally dominant, and whatever tuning is applied it by necessity has to include also off-diagonal Hamiltonian matrix elements. In this work, we show how the problems related to off-diagonal matrix elements can be circumvented by a scheme that transforms from *jj*-coupling to *LSJ*-coupling and then back again.

## 2. Multiconfiguration Methods

*Expansions over Configuration State Functions in LSJ- and jj-Coupling*

In multiconfiguration methods, the wave function of an atomic state $\Gamma J M_J$, with $\Gamma$ being its identifying label, $J$ the total angular momentum quantum number, and $M_J$ the total magnetic quantum number, is approximated by an atomic state function (ASF), which is a linear combination of CSFs

$$\Psi(\Gamma J M_J) = \sum_{i=1}^{N_{CSF}} c_i \Phi(\gamma_i J M_J), \tag{1}$$

where $\gamma_i$ specifies the orbital occupancy and spin-angular coupling tree quantum numbers of each CSF. Requiring the energy computed from the multiconfiguration expansion to be stationary with respect to perturbations in the expansion coefficients leads to a matrix eigenvalue problem,

$$(\boldsymbol{H} - E\boldsymbol{I})\boldsymbol{c}^{\mathrm{T}} = \boldsymbol{0}, \tag{2}$$

where $\boldsymbol{c} = (c_1, c_2, \ldots, c_M)$ is the vector of expansion coefficients and $\boldsymbol{H}$ is the $M \times M$ Hamiltonian matrix with elements $H_{ij} = \langle \Phi(\gamma_i J M_J) | \mathcal{H} | \Phi(\gamma_j J M_J) \rangle$ of the appropriate Hamiltonian operator $\mathcal{H}$. The latter depends on the formalism: the non-relativistic formalism with corrections in the Breit–Pauli approximation or the fully relativistic Dirac formalism with the transverse photon interaction [5,11–13].

The CSFs in the non-relativistic formalism, with relativistic corrections in the Breit–Pauli approximation, are constructed from a set of spin-orbitals

$$\psi_{nlm_lm_s}(r, \theta, \varphi) = \frac{P_{nl}(r)}{r} Y_{lm_l}(\theta, \varphi) \chi_{m_s}^{(1/2)},$$

where $P_{nl}(r)$ is the radial function, $Y_{lm_l}(\theta, \varphi)$ a spherical harmonic and $\chi_{m_s}^{(1/2)}$ a spin function. A general non-relativistic configuration consists of $m$ groups of equivalent electrons

$$(n_1l_1)^{w_1}(n_2l_2)^{w_2}\ldots(n_ml_m)^{w_m}, \quad N = \sum_{i=1}^{m} w_i,$$

where $w_i$ is the occupation number of the subshell $i$ and $N$ the total number of electrons. Antisymmetric and *LS*-coupled functions for each group of equivalent electrons are built from products of spin-orbitals using recursive methods in terms of coefficients of fractional parentage (CFPs). To construct the CSFs associated with the above configuration, one starts with the products of the antisymmetric eigenfunctions for the different groups of equivalent electrons. With the repeated use of vector coupling expansions, we can couple the product functions to the final total angular momenta $LM_LSM_S$. Coupling applies from left to right and $L_1S_1$ of the first group of equivalent electrons are coupled to $L_2S_2$ of the second group to intermediate angular momenta $L_{12}S_{12}$, which in turn is coupled to $L_3S_3$ of the third group and so on until we obtain the final total angular momenta $LS$

$$(\ldots((L_1S_1L_2S_2)L_{12}S_{12}L_3S_3)L_{123}S_{123}\ldots)LM_LSM_S. \tag{3}$$

This procedure leads to functions which are antisymmetric with respect to co-ordinate permutations within each subshell, but not antisymmetric with respect to permutations between different subshells [14]. The additional antisymmetrization can, however, be accomplished through the restricted permutations involving co-ordinate exchange only between two different subshells such that the co-ordinate number within each subshell remains in an increasing order. As a final step, the $L$ and $S$ are coupled to a final $J$. This coupling we refer to as $LSJ$-coupling. We can now write a CSF associated with the configuration as

$$\Phi((n_1l_1)^{w_1}\alpha_1\nu_1(n_2l_2)^{w_2}\alpha_2\nu_2\, L_{12}S_{12}(n_3l_3)^{w_3}\alpha_3\nu_3\, L_{123}S_{123}\ldots(n_ml_m)^{w_m}\alpha_m\nu_m\, LS\, JM_J), \quad (4)$$

where $\alpha$ and $\nu$ are additional quantum numbers needed to obtain a one-to-one classification of the energy levels. The construction of the $LSJ$-coupled CSFs is detailed in [11].

The CSFs in the relativistic formalism are constructed from a set of Dirac orbitals

$$\psi_{nlsjm}(r,\theta,\varphi) = \frac{1}{r}\left(\begin{array}{c} P_{nlj}(r)\,\Omega_{lsjm_j}(\theta,\varphi) \\ \mathrm{i}\,Q_{nlj}(r)\,\Omega_{\tilde{l}sjm_j}(\theta,\varphi) \end{array}\right), \quad (5)$$

where $P_{nlj}(r)$ and $Q_{nlj}(r)$ are the radial functions and $\Omega_{lsjm_j}(\theta,\varphi)$ are two-component spherical spinors built from the coupling of the spherical harmonics $Y_{lm_l}(\theta,\varphi)$ and the spin functions $\chi_{m_s}^{(1/2)}$. A general relativistic configuration consists of $\widetilde{m}$ groups of equivalent electrons

$$(n_1l_1j_1)^{w_1}(n_2l_2j_2)^{w_2}\ldots(n_{\widetilde{m}}l_{\widetilde{m}}j_{\widetilde{m}})^{w_{\widetilde{m}}}, \quad N = \sum_{i=1}^{\widetilde{m}} w_i, \quad (6)$$

where $w_i$ is the occupation number of the relativistic subshell $i$ and $N$ the total number of electrons. The construction follows the prescription in the non-relativistic case. Antisymmetric and $jj$-coupled functions for each group of equivalent electrons are built from products of spin-orbitals using recursive methods in terms of CFPs. To construct the CSFs associated with the above relativistic configuration, one starts with the products of the antisymmetric eigenfunctions for the different groups of equivalent electrons. With the repeated use of vector coupling expansions, we can couple the product functions to the final total angular momenta $JM_J$. Coupling applies from left-to-right and $J_1$ of the first group of equivalent electrons are coupled to $J_2$ of the second group to intermediate angular momenta $J_{12}$, which in turn is coupled to $J_3$ of the third group and so on until we obtain the final total angular momenta $JM_J$

$$(\ldots((J_1J_2)J_{12}J_3)J_{123}\ldots)JM_J. \quad (7)$$

This coupling we refer to as $jj$-coupling. Performing restricted permutations involving co-ordinate exchange only between two different relativistic subshells, we end up with the most general form of a CSF

$$\Phi((n_1l_1j_1)^{w_1}\alpha_1\nu_1J_1\,(n_2l_2j_2)^{w_2}\alpha_2\nu_2J_2\,J_{12}(n_3l_3j_3)^{w_3}\alpha_3\nu_3J_3\,J_{123}\ldots(n_{\widetilde{m}}l_{\widetilde{m}}j_{\widetilde{m}})^{w_{\widetilde{m}}}\alpha_{\widetilde{m}}\nu_{\widetilde{m}}J_{\widetilde{m}}\,JM_J). \quad (8)$$

The CSFs in $LSJ$- and $jj$-couplings belonging to a number of configurations constitute two different sets of orthonormal basis functions. In the non-relativistic limit,

$$P_{nlj}(r) \rightarrow P_{nl}(r), \quad \text{and} \quad Q_{nlj}(r) \rightarrow 0, \quad (9)$$

they span the same vector space.

## 3. Hamiltonian Matrix in $LSJ$- and $jj$-Coupling

For light systems, where the relativistic effects are small, the CSFs in $LSJ$-coupling give a Hamiltonian with small off-diagonal elements between CSFs with different $LS$. As a

consequence, the CSF expansion for a state will be dominated by CSFs with the same *LS* quantum numbers. These quantum numbers are to some approximation 'good' quantum numbers. The situation is very different in *jj*-coupling. There are no approximate good quantum numbers, and there are in many cases large off-diagonal elements between CSFs.

We take the $2s2p\,^3P_1$ and $2s2p\,^1P_1$ states in C III as an illustrative example. In the non-relativistic formalism, with relativistic corrections in the Breit–Pauli approximation, the two states can, in the simplest approximation, be described by two CSFs (where we have suppressed the $M_J$ quantum number for brevity)

$$\Phi_1 = |2s2p\,^3P_1\rangle \quad \text{and} \quad \Phi_2 = |2s2p\,^1P_1\rangle. \tag{10}$$

Given radial orbitals from an *LS* averaged Hartree–Fock (HF) calculation and running the Breit–Pauli configuration interaction (CI) program of the ATSP2K package [10] we obtain

$$\boldsymbol{H} = \begin{pmatrix} -36.248683 & 0.000184 \\ 0.000184 & -35.970479 \end{pmatrix}. \tag{11}$$

The off-diagonal matrix elements between the CSFs with different *LS* terms are small compared with the difference ($36.248683 - 35.970479 = 0.278204$) between the diagonal matrix elements. Diagonalizing the Hamiltonian matrix gives the energies

$$E_1 = -36.248683\ \text{E}_h \quad \text{and} \quad E_2 = -35.970479\ \text{E}_h \tag{12}$$

and the eigenvectors

$$\Psi(\text{“}2s2p\,^3P_1\text{”}) = -0.99999978\,\Phi_1 + 0.00066201\,\Phi_2 \tag{13}$$

and

$$\Psi(\text{“}2s2p\,^1P_1\text{”}) = -0.00066201\,\Phi_1 - 0.99999978\,\Phi_2, \tag{14}$$

where the use of quotation marks for the labels to the left highlights the fact that the notation is just an identifying label, even though the wave function is a mixture of CSFs.

In the relativistic formalism the two states are, again in the simplest approximation, described by

$$\Phi_1 = |(2s_{1/2}2p_{1/2})_1\rangle \quad \text{and} \quad \Phi_2 = |(2s_{1/2}2p_{3/2})_1\rangle. \tag{15}$$

Determining, to be as consistent as possible with the calculation above, the relativistic orbitals in the Pauli limit [15,16]

$$P_{nlj}(r) = P_{nl}^{HF}(r), \quad Q_{nlj}(r) = \frac{\alpha}{2}\left(\frac{d}{dr} + \frac{\kappa}{r}\right)P_{nl}^{HF}(r) \tag{16}$$

where

$$\kappa = \begin{cases} -(l+1) & \text{for } j = l + 1/2 \\ +l & \text{for } j = l - 1/2 \end{cases} \tag{17}$$

and running the CI program of the fully relativistic GRASP2018 package [17] including the Breit interaction yields the following Hamiltonian

$$\boldsymbol{H} = \begin{pmatrix} -36.063028 & 0.131089 \\ 0.131089 & -36.156113 \end{pmatrix}. \tag{18}$$

Now the off-diagonal matrix elements between the CSFs are of the same magnitude as the difference ($36.156113 - 36.063028 = 0.09308$) between the diagonal matrix elements. Diagonalizing the Hamiltonian gives the energies

$$E_1 = -36.248677\ \text{E}_h \quad \text{and} \quad E_2 = -35.970464\ \text{E}_h \tag{19}$$

and the eigenvectors

$$\Phi(\text{"}2s2p\,^3P_1\text{"}) = 0.81687864\,\Phi_1 - 0.57680957\,\Phi_2 \tag{20}$$

and

$$\Phi(\text{"}2s2p\,^1P_1\text{"}) = 0.57680957\,\Phi_1 + 0.81687864\,\Phi_2. \tag{21}$$

The energies are essentially the same as for the MCHF and Breit–Pauli calculation. The small differences can be attributed to differences in the order to which relativistic effects are included. The eigenvector composition, however, is totally different from the *LSJ*-coupling case.

## 4. Transition Parameters in *LSJ*- and *jj*-Coupling

The differences in eigenvector composition between the *LSJ*- and *jj*-coupling schemes have a huge impact on the computation of transition rates. For simplicity, we restrict the discussion to electric dipole (E1) transitions in the long wavelength approximation. The different transition parameters, line strength, transition rates, oscillator strengths, for a transition from an upper state $\Gamma'J'M_{J'}$ to any of the $2J + 1$ states $\Gamma J M_J$, $M_J = -J, -J + 1, \ldots, J$ of a lower energy level are proportional to the line strength

$$S(\Gamma'J', \Gamma J) = |\langle \Psi(\Gamma'J') \| \mathbf{D}^{(1)} \| \Psi(\Gamma J) \rangle|^2, \tag{22}$$

where $\mathbf{D}^{(1)}$ is the dipole operator in length or velocity gauge (Babushkin and Coulomb gauge in relativistic theory), see [5,13,18]. The factors of proportionality are mixtures of powers of the transition energy $\Delta E$, statistical factors and physical constants, depending on the type of the parameter and gauge.

The dynamical range of E1 transitions is huge, and the transitions are most often sorted in different categories. *LS*-allowed transitions are normally strong and occur between states for which the dominant components in *LSJ*-coupling fulfill the selection rules. *LS*-forbidden transitions, or intercombination transitions, are often weak and occur between states for which the dominant component in *LSJ*-coupling breaks one of the above selection rules. If $\Delta S = 1$ we talk about a spin-forbidden transition.

To fully appreciate the differences between the computations of transition rates in *LSJ*- and *jj*-coupling schemes, we look at the line strength in the length (Babushkin) gauge for the $2s^2\,^1S_0 - 2s2p\,^3P_1$ intercombination transition in C III. Approximating the $2s^2\,^1S_0$ ground state with a single CSF, $|2s^2\,^1S_0\rangle$, and using the expansion given in Equation (13)

$$\Psi(\text{"}2s2p\,^3P_1\text{"}) = -0.99999978\,|2s2p\,^3P_1\rangle + 0.00066201\,|2s2p\,^1P_1\rangle \tag{23}$$

we have

$$S(^1S_0,\,^3P_1) = |0.00066201\,\langle 2s^2\,^1S_0 \| \mathbf{D}^{(1)} \| 2s2p\,^1P_1 \rangle|^2\,, \tag{24}$$

where we have used that the electric dipole transition matrix element between states with different spins is zero. The reduced transition matrix element between the singlet states is

$$\langle 2s^2\,^1S_0 \| \mathbf{D}^{(1)} \| 2s2p\,^1P_1 \rangle = 1.873529, \tag{25}$$

which yields $S(^1S_0,\,^3P_1) = 1.538332 \times 10^{-6}$. From Equation (24) it is now clear the smallness of the line strength comes from the smallness of the $^1P_1$ mixing into the wave function of the $2s2p\,^3P_1$ state. The situation is very different in *jj*-coupling; see [19] for an excellent discussion. Now

$$\Psi(\text{"}2s2p\,^3P_1\text{"}) = 0.81687864\,|(2s_{1/2}2p_{1/2})_1\rangle - 0.57680957\,|(2s_{1/2}2p_{3/2})_1\rangle \tag{26}$$

and we have

$$S(^1S_0, {}^3P_1) = \quad |0.81687864 \langle (2s_{1/2}^2)_0 \| \mathbf{D}^{(1)} \| (2s_{1/2}2p_{1/2})_1 \rangle \tag{27}$$
$$-0.57680957 \langle (2s_{1/2}^2)_0 \| \mathbf{D}^{(1)} \| (2s_{1/2}2p_{3/2})_1 \rangle|^2.$$

The reduced transition matrix elements are

$$\langle (2s_{1/2}^2)_0 \| \mathbf{D}^{(1)} \| (2s_{1/2}2p_{1/2})_1 \rangle = 1.081695, \quad \langle (2s_{1/2}^2)_0 \| \mathbf{D}^{(1)} \| (2s_{1/2}2p_{3/2})_1 \rangle = 1.529757 \tag{28}$$

which gives $S(^1S_0, {}^3P_1) = 1.525381 \times 10^{-6}$, in perfect agreement with the value above. In this case, the smallness of the line strength comes from subtraction of two large, but almost cancelling, contributions. We have a numerically ill-conditioned problem with a cancellation of three orders of magnitude. Even a small change of the mixing coefficients, due to added electron correlation or fine-tuning, may have an appreciable effect on the line strength.

## 5. Transformation between Coupling Schemes

We denote the CSF basis in $jj$-coupling belonging to one or more relativistic configurations by $\mathbf{\Phi} = (\Phi_1, \Phi_2, \dots, \Phi_m)^{\mathrm{T}}$ and the corresponding CSF basis in $LSJ$-coupling by $\widehat{\mathbf{\Phi}} = (\widehat{\Phi}_1, \widehat{\Phi}_2, \dots, \widehat{\Phi}_m)^{\mathrm{T}}$. The two bases are related through

$$\widehat{\mathbf{\Phi}} = T_{LSJ,jj}^{\mathrm{T}} \mathbf{\Phi}, \tag{29}$$

where $T_{LSJ,jj}$ is the coupling transformation matrix. For two-electron systems, there are simple analytical expressions for the transformation matrix; see ([20], p. 249). In the general many-electron case, the transformation matrix is computed with a modified version of the jj2lsj program [21] in which $T_{LSJ,jj} \equiv \langle \gamma_s JP \| \gamma_r L_r S_r JP \rangle$ ([22], Equation (5)) is expressed in terms of the transformation between the $jj$- and $LSJ$-couplings of the subshells and additionally taking into account the transformations between the $jj$- and $LSJ$-couplings inside the subshell states $\langle l^w \alpha v LSJ | (l_-^{w_1} v_1 J_1, l_+^{(w-w_1)} v_2 J_2) J \rangle$ with the same $l$ orbital quantum number ([21], Equation (3)). For more information about the transformation, see [23–26].

An atomic state function $\Psi$ can be expressed in both the $\Phi$ base and the $\widehat{\Phi}$ base

$$\Psi = c_1 \Phi_1 + c_2 \Phi_2 + \dots + c_m \Phi_m = \widehat{c}_1 \widehat{\Phi}_1 + \widehat{c}_2 \widehat{\Phi}_2 + \dots + \widehat{c}_m \widehat{\Phi}_m. \tag{30}$$

Given the transformation matrix, the vector $c$ with expansion coefficients in the $jj$-coupled basis is transformed to a vector $\widehat{c}$ in the $LSJ$-coupled basis according to

$$\widehat{c} = T_{LSJ,jj}^{-1} c. \tag{31}$$

Using the fact that the transformation between two orthonormal bases is unitary, we have $T_{LSJ,jj}^{-1} = T_{LSJ,jj}^{\mathrm{T}}$ and

$$\widehat{c} = T_{LSJ,jj}^{\mathrm{T}} c. \tag{32}$$

The Hamiltonian matrix $H$ in the $jj$-coupled basis with elements $H_{ij} = \langle \Phi_i | \mathcal{H} | \Phi_j \rangle$ is transformed to a matrix $\widehat{H}$ in the $LSJ$-coupled basis with elements $\widehat{H}_{ij} = \langle \widehat{\Phi}_i | \mathcal{H} | \widehat{\Phi}_j \rangle$ according to

$$\widehat{H} = T_{LSJ,jj}^{-1} H \, T_{LSJ,jj}. \tag{33}$$

With the use of $T_{LSJ,jj}^{-1} = T_{LSJ,jj}^{\mathrm{T}}$ this can also be written

$$\widehat{H} = T_{LSJ,jj}^{\mathrm{T}} H \, T_{LSJ,jj}. \tag{34}$$

The $jj$- to $LSJ$-coupling transformation changes the wave function representation and also the Hamiltonian matrix. The wave functions themselves, along with computed properties, remain invariant.



## 6. Fine-Tuning of Eigenvalues

All atomic calculations are approximate. Depending on the scale of the calculation (number of CSFs, size of orbital set and the amount of included correlation effects) and complexity of the atomic system, the observed excitation or transition energies may be predicted more or less accurately. Known problematic cases are transition energies between states with different total spins. States where the spins of the outer electrons line up to give maximal spin quantum numbers are in general associated with less electron correlation energy than states where they do not line up. This often leads to cases where the energies of high spin states are too low in comparison with the energies of low spin states ([5], pp. 96–97). Other known problematic cases are perturber states in Rydberg sequences, where, depending on the case, the perturber state may be too low or too high in relation to the Rydberg states [7,8]. It is in general very hard to optimize a balanced orbital set that accurately positions perturber states in the often rather densely spaced Rydberg series. A failure to accurately predict the energy structure most often translates to uncertain transition rates.

Fine-tuning is a semi-empirical correction process that is applied to the diagonal elements of the $M \times M$ Hamiltonian matrix in Equation (2) in order to rectify the CSF expansion coefficients and improve computed energy separations and transition parameters. Let, closely following Hibbert [3], $\Phi_1$ and $\Phi_2$ be single CSFs or CSF expansions approximately describing two states. We now want to add another CSF $\Phi_3$ to improve the representation of the second state, where, for simplicity, we assume $\langle \Phi_1 | \mathcal{H} | \Phi_3 \rangle = 0$. This gives a Hamiltonian of the form

$$\begin{pmatrix} A & h & 0 \\ h & B & f \\ 0 & f & C \end{pmatrix} \tag{35}$$

where, as a consequence of the fact that $\Phi_1$ and $\Phi_2$ approximately describes the two states, $|h| \ll |A - B|$. We now diagonalize the $2 \times 2$ submatrix associated with the $\Phi_2$ and $\Phi_3$ interaction to obtain the diagonal matrix

$$\begin{pmatrix} B' & 0 \\ 0 & C' \end{pmatrix} \tag{36}$$

and the eigenvectors

$$\Phi_2' = \alpha \Phi_2 + \beta \Phi_3 \quad \text{and} \quad \Phi_3' = -\beta \Phi_2 + \alpha \Phi_3 \tag{37}$$

with $\alpha^2 + \beta^2 = 1$. Typically, $|f| \ll |B - C|$, which, to a good approximation, gives the eigenvalues

$$B' = B + \frac{f^2}{B - C} \quad \text{and} \quad C' = C - \frac{f^2}{B - C}. \tag{38}$$

The coefficients of the eigenfunctions are, again to a good approximation, given by

$$\alpha = \frac{1}{\sqrt{1 + x^2}} \quad \text{and} \quad \beta = \alpha x = \frac{x}{\sqrt{1 + x^2}}, \tag{39}$$

where $x = f / |B - C|$. In the $\Phi_1, \Phi_2', \Phi_3'$ basis, the Hamiltonian becomes

$$\begin{pmatrix} A & q & p \\ q & B' & 0 \\ p & 0 & C' \end{pmatrix} \tag{40}$$

with

$$p = \langle \Phi_1 | \mathcal{H} | \Phi_3' \rangle = -\beta h = -\alpha h x \quad \text{and} \quad q = \langle \Phi_1 | \mathcal{H} | \Phi_2' \rangle = \alpha h. \tag{41}$$

Since $x \ll 1$ we infer that $\alpha \approx 1$, $q \approx h$ and $|p| \ll |q|$. The major effect of the interaction with $\Phi_3$ is thus a change of the original $2 \times 2$ submatrix associated with the $\Phi_1$ and $\Phi_2$ interaction

$$\begin{pmatrix} A & h \\ h & B \end{pmatrix} \rightarrow \begin{pmatrix} A & q \\ q & B' \end{pmatrix}. \tag{42}$$

We now understand the mechanism by which fine-tuning works: the inclusion of an extra CSF $\Phi_3$ can instead be effected to a good approximation by a change to the diagonal matrix element $B$. In the general context, aiming for several states, and where we want to correct their relative positions, we tune the diagonal matrix elements in the Hamiltonian matrix. The requirements are that the off-diagonal matrix elements are small relative to the differences in the diagonal matrix elements.

## 7. Fine-Tuning of Eigenvalues in *LSJ*-Coupling

We apply the fine-tuning process to the $2s2p\ ^3P_1$ and $2s2p\ ^1P_1$ states in C III. The Hamiltonian matrix for the two *LSJ*-coupled CSFs in the Breit–Pauli approximation is given in Equation (11). As discussed in the introduction, there is more electron correlation in the $2s2p\ ^1P_1$ state than in $2s2p\ ^3P_1$ and in the limited two CSF calculation the $2s2p\ ^1P_1$ state is far too high relative to $2s2p\ ^3P_1$ ($\Delta E = E_2 - E_1 = 61{,}055.99$ cm$^{-1}$ compared with $\Delta E = 49{,}961.29$ cm$^{-1}$ from NIST). To bring the $2s2p\ ^1P_1$ state down, we subtract $61{,}056 - 49{,}961 = 11{,}095$ cm$^{-1}$ from the corresponding diagonal matrix element to yield

$$H = \begin{pmatrix} -36.248683 & 0.000184 \\ 0.000184 & -36.021031 \end{pmatrix}. \tag{43}$$

Diagonalizing the Hamiltonian matrix gives the energies

$$E_1 = -36.248683\ \mathrm{E}_h \quad \text{and} \quad E_2 = -36.021031\ \mathrm{E}_h. \tag{44}$$

The energy difference is $\Delta E = E_2 - E_1 = 49{,}961.51$ cm$^{-1}$, in good agreement with what we aimed for. The change in the diagonal matrix element leads to modified eigenvalue compositions

$$\Psi(\text{"}2s2p\ ^3P\text{"}, J = 1) = -0.99999967\ \Phi_1 + 0.00080901\ \Phi_2 \tag{45}$$

and

$$\Psi(\text{"}2s2p\ ^1P\text{"}, J = 1) = -0.00080901\ \Phi_1 - 0.99999967\ \Phi_2. \tag{46}$$

Fine-tuning affects the transition parameters in two ways: firstly, the fine-tuned energy modifies the energy factor $\Delta E$ that is multiplied with the line strength to obtain, e.g., transition rates and oscillator strengths, and secondly, the line strength changes due to the fact that the mixing coefficients change.

## 8. Fine-Tuning of Eigenvalues in *jj*-Coupling

Fine-tuning is not directly applicable in the *jj*-coupled case due to the fact that a state is not well described by a single CSF; it is a combination of two or more CSFs that describes the state. This means that we have large off-diagonal matrix elements, and a tuning of the diagonal matrix elements will not result in the desired energy separations. To circumvent the problems of fine-tuning in *jj*-coupling, we have developed a method in which the Hamiltonian in the *jj*-coupled basis is transformed to a matrix in the corresponding *LSJ*-coupled basis, for which now the prerequisites for fine-tuning often, but not always, are fulfilled. We fine-tune by modifying the diagonal matrix elements and transform back to a matrix in the *jj*-coupled basis. This in effect modifies both diagonal and off-diagonal Hamiltonian matrix elements. The procedure can be summarized as follows:

- arrange the full CSF expansion so that the CSFs in the multireference (MR) come first.
- compute the coupling transformation matrix $T_{LSJ,jj}$ between the $jj$-coupled CSFs in the MR and the corresponding $LSJ$-coupled CSFs.
- perform a relativistic CI calculation for the full CSF expansion. Save the Hamiltonian on disk in sparse format.
- read and transform the Hamiltonian submatrix corresponding to the CSFs in the MR from $jj$-coupling ($H_{jj}$) to $LSJ$-coupling ($H_{LSJ}$) according to

$$H_{LSJ} = T_{LSJ,jj}^{\mathrm{T}} \, H_{jj} \, T_{LSJ,jj}. \tag{47}$$

- allow the user to fine-tune the diagonal elements of $H_{LSJ}$ to yield $H_{LSJ}^{\mathrm{ft}}$.
- transform the fine-tuned Hamiltonian matrix $H_{LSJ}^{\mathrm{ft}}$ back to $jj$-coupling according to

$$H_{jj}^{\mathrm{ft}} = T_{LSJ,jj} \, H_{LSJ}^{\mathrm{ft}} \, T_{LSJ,jj}^{\mathrm{T}} \tag{48}$$

  and merge it into the Hamiltonian saved on disk.
- perform a relativistic CI calculation for the full CSF expansion based on the Hamiltonian for which the submatrix corresponding to the CSFs in the MR was modified.

## 9. Program Implementation

The procedure for fine-tuning in $jj$-coupling is implemented in the GRASP2018 package [17] through the new programs `jj2lsj_2022` and `rfinetune`, both publically available at the GRASP GitHub repository at https://github.com/compas, accessed on 27 March 2023. The programs assume that an `rci` calculation has been performed for a CSF expansion (list), where the CSFs in the MR appear first in the list, so that a restart file, `rci.res`, containing the Hamiltonian matrix in sparse format is available, (see [27], Section 6.7). If an MPI run has been performed using the `rci_mpi` program, the Hamiltonian is distributed, and in this case there is one restart file per processor residing in the directories defined by the `disks` file, see ([27], Section 6.4). In addition, the CSFs in the MR should be available in a separate list. The `jj2lsj_2022` program reads the $jj$-coupled CSFs in the MR list and generates the corresponding $LSJ$-coupled CSFs that are saved to file. In addition, the program computes the transformation matrix between the $jj$-coupled CSFs in the MR and the corresponding $LSJ$-coupled CSFs and writes it to a binary file `.lsj.T`. The `rfinetune` program reads the list of $LSJ$-coupled CSFs, the transformation matrix and the Hamiltonian submatrix corresponding to the CSFs in the MR from the `rci.res` file. The Hamiltonian submatrix is transformed from $jj$- to $LSJ$-coupling. For each $LSJ$-coupled CSF, the user has the opportunity to fine-tune the corresponding diagonal matrix element of the transformed Hamiltonian with a prescribed energy that is entered in cm$^{-1}$. After the tuning, the program transforms the Hamiltonian submatrix back to $jj$-coupling and inserts it into the `rci.res` file. Finally, running the `rci` (or `rci_mpi`) program in restart mode, so that all matrix elements are read from the updated `rci.res` file, yields the fine-tuned energies and the modified expansion coefficients of the CSFs in the full list.

## 10. Test-Run

We test the new programs on the $2s^2 \, {}^1S_0$, $2s2p \, {}^1P_1$ and $2s2p \, {}^3P_{0,1,2}$ states in C III and aim to fine-tune the submatrix corresponding to the MR of the odd states. We save, for future use, the MR list of the odd states in a file `DF_odd.c`. The file is shown below.

```
Core subshells:

Peel subshells:
  1s   2s   2p-  2p
CSF(s):
  1s ( 2)  2s ( 1)  2p-( 1)
              1/2       1/2
```

```
                                  0-
 *
  1s ( 2)   2s ( 1)   2p ( 1)
                1/2       3/2
                                  1-
  1s ( 2)   2s ( 1)   2p-( 1)
                1/2       1/2
                                  1-
 *
  1s ( 2)   2s ( 1)   2p ( 1)
                1/2       3/2
                                  2-
```

Some preparatory work is needed, and we start by generating a set of radial orbitals up to $n = 7$ ($\{7s, 7p, \ldots, 7h, 7i\}$) in extended optimal level (EOL) MCDHF calculations based on CSF expansions accounting for valence–valence electron correlation of the above even and odd states. The MCDHF calculations are followed by an `rci_mpi` calculation using, in this test-run, four processors for the even $2s^2\,^1S_0$ state accounting for the Breit interaction. The CSF list and the wave function file for the $2s^2\,^1S_0$ state are saved in `2s2.c` and `2s2.w` and the `rci_mpi` calculation outputs the `2s2.cm` mixing file. The `rci_mpi` calculation for the even state is followed by an `rci_mpi` calculation, again using four processors, for the odd $2s2p\,^1P_1$, $2s2p\,^3P_{0,1,2}$ states accounting for the Breit interaction. The CSF list and the wave function file for the odd states are saved in `2s2p.c` and `2s2p.w` and the `rci_mpi` calculation outputs the `2s2p.cm` mixing file. The `rci_mpi` calculation for the odd states using four processors also gives four restart files `rci000.res`, `rci001.res`, `rci002.res`, `rci003.res` containing the Hamiltonian matrix in sparse format. The files reside in the directories set by the `disks` file. For the `rci_mpi` calculation of the odd states we used the `rcsfzerofirst` program to ensure that the CSFs in the MR came first in the list, see ([27], Section 14.1). The excitation energies from the `rci_mpi` calculations are shown in Table 1 together with the experimental energies from the NIST the database [28].

**Table 1.** Excitation energies from `rci_mpi` calculations compared with experimental energies from the NIST database. The differences give how much the energies of the odd states should be fine-tuned to be in accordance with the experimental energies.

| No | Pos | J | Parity | Configuration | RCI (cm^-1) | NIST (cm^-1) | fine-tune (cm^-1) |
|----|-----|---|--------|---------------|-------------|--------------|-------------------|
| 1 | 1 | 0 | + | 2s(2)_1S | 0.00 | 0.00 | |
| 2 | 1 | 0 | - | 2s_2S.2p_3P | 52463.43 | 52367.06 | -96 |
| 3 | 1 | 1 | - | 2s_2S.2p_3P | 52486.91 | 52390.75 | -96 |
| 4 | 1 | 2 | - | 2s_2S.2p_3P | 52543.19 | 52447.11 | -96 |
| 5 | 2 | 1 | - | 2s_2S.2p_1P | 102530.94 | 102352.04 | -179 |

The last column gives how much the energies of the odd states should be fine-tuned to be in accordance with the experimental energies. The main observation is that the $2s2p\,^3P$ states are somewhat too high compared with the $2s^2\,^1S_0$ ground state and should be pushed down. The $2s2p\,^1P_1$ state is also somewhat too high and should be pushed down.

We are now in the position to fine-tune and start by running `jj2lsj_2022` for the four CSFs in the MR saved in `DF_odd.c` to obtain the corresponding $LSJ$-coupled CSFs along with the transformation matrix.

```
     **********************************************************************
     *    RUN JJ2LSJ_2022 TO OBTAIN TRANSFORMATION MATRIX                 *
     *    AND LSJ-COUPLING CSFs LIST                                      *
     *    INPUT FILES: name.c, name.(c)m                                  *
     *      (optional) name.lsj.T                                         *
     *    OUTPUT FILES: name.lsj.lbl                                      *
     *      (optional)  name.lsj.c, name.lsj.j,name.uni.lsj.lbl,          *
     *                  name.uni.lsj.sum, name.lsj.T                      *
     **********************************************************************

     >>jj2lsj_2022

      jj2lsj: Transformation of ASFs from a jj-coupled CSF basis
              into an LS-coupled CSF basis  (Fortran 95 version)
              (C) Copyright by   G. Gaigalas and Ch. F. Fischer,
              (2022).
              Input files: name.c, name.(c)m
               (optional)  name.lsj.T
              Ouput files: name.lsj.lbl,
               (optional)  name.lsj.c, name.lsj.j,
                           name.uni.lsj.lbl, name.uni.lsj.sum,
                           name.lsj.T

      Name of state
     >>DF_odd
      Loading Configuration Symmetry List File ...
      There are 4 relativistic subshells;
      There are 4 relativistic CSFs;
       ... load complete;

      Mixing coefficients from a CI calc.?
     >>y
      Do you need a unique labeling? (y/n)
     >>n
         nelec  =           4
         ncftot =           4
         nw     =           4
         nblock =           3

        block      ncf      nev    2j+1  parity
           1        1        1       1      -1
           2        2        2       3      -1
           3        1        1       5      -1
      Default settings?  (y/n)
     >>n
      All levels (Y/N)
     >>y
      Maximum % of omitted composition
     >>0.0
      What is the value below which an eigenvector composition
      is to be neglected for printing?
     >>0.0
      Do you need the transformation output file *.lsj.T?  (y/n)
     >>y
```

```
Below  1.0E-16 the eigenvector component is to be neglected for
calculating Below  0.0E+00 the eigenvector composition is to be
neglected for~printing

 ...........

Finish Date and Time:
  Date (Yr/Mon/Day): 2022/02/01
  Time (Hr/Min/Sec): 20/34/08.044
  Zone: +0800

jj2lsj: Execution complete.
```

The four *LSJ*-coupled CSFs generated by jj2lsj_2022 are available in the file DF_odd.lsj.c, which is displayed below.

```
 1s( 2)  2s( 1)  2p( 1)
1S0 2S1 2P1 2S   3P
*
 1s( 2)  2s( 1)  2p( 1)
1S0 2S1 2P1 2S   1P
 1s( 2)  2s( 1)  2p( 1)
1S0 2S1 2P1 2S   3P
*
 1s( 2)  2s( 1)  2p( 1)
1S0 2S1 2P1 2S   3P
*
```

We now run the rfinetune program. The program reads the list of *LSJ*-coupled CSFs, the transformation matrix and the Hamiltonian submatrix corresponding to the CSFs in the MR from the rci000.res, rci001.res, rci1002.res and rci003.res files. The Hamiltonian submatrix is transformed from *jj*-coupling to to *LSJ*-coupling. For each *LSJ*-coupled CSF, the user has the opportunity to fine-tune the corresponding diagonal matrix element of the transformed Hamiltonian with a prescribed energy.

```
**********************************************************************
*     RUN RFINETUNE TO FINE-TUNE THE HAMILTONIAN MATRIX FROM        *
*              THE RCI_MPI RUN ON FOUR PROCESSORS                   *
*     INPUT FILES: rci000.res, rci001.res, rci002.res, rci003.res, *
*              DF_odd.lsj.T,DF_odd.lsj.T,DF_odd.c                   *
*     OUTPUT FILE: rci000.resnew, rci001.resnew,                   *
*              rci002.resnew, rci003.resnew                        *
**********************************************************************

>>rfinetune

 RFINETUNE
 This is the rfinetune program
 Input files:  rci.res, name.lsj.T, name.lsj.c,name.c
 Output files: rci.resnew

 Transformation matrix is from calculation of:
 0--serial
 1--parallel
>>1
```

```
 Name of MR state:
>>DF_odd
 rci.res is from parallel calculation...
 Name of the temporary directory: (e.g.'/home/user/tmp/')
>>'/home/ytli/tmp/'
 There are           4 files in the temp directory...

 BLOCK           1
 No. in LSJ-couping =           1
   1s( 2)  2s( 1)  2p( 1)
  1S0 2S1 2P1 2S  3P
 How many diagonal elements should be finetuned:
>>1
 Give serial the number of the CSF in LSJ-couping you should
 fine-tune together with the energy change in cm-1
>>1,-96

 BLOCK           2
 No. in LSJ-couping =           1
   1s( 2)  2s( 1)  2p( 1)
  1S0 2S1 2P1 2S  1P
 No. in LSJ-couping =           2
   1s( 2)  2s( 1)  2p( 1)
  1S0 2S1 2P1 2S  3P
 How many diagonal elements should be fine-tuned:
>>2
 Give the serial number of the CSF in LSJ-couping you should
 fine-tune together with the energy change in cm-1
>>1,-179
 Give the serial number of the CSF in LSJ-couping you should
 fine-tune  together with the energy change in cm-1
>>2,-96

 BLOCK           3
 No. in LSJ-couping =           1
   1s( 2)  2s( 1)  2p( 1)
  1S0 2S1 2P1 2S  3P
 How many diagonal elements should be fine-tuned:
>>1
 Give serial the number of the CSF in LSJ-couping you should
 fine-tune together with the energy change in cm-1
>>1,-96
 Created rcixxx.resnew in /home/ytli/tmp/
```

The modified matrix elements corresponding to the CSFs in the MR are merged into the full set of matrix elements and written to `rci000.resnew`, `rci001.resnew`, `rc1002.resnew` and `rci003.resnew`. These files should be renamed before running `rci_mpi` in restart mode.

```
**********************************************************************
*    RENAME ALL THE FINE-TUNED RESTART FILES ASSUMING THAT THE      *
*    RESTART FILES RESIDE IN /home/ytli/tmp/                        *
*    CHECK THE DISKS FILE IN YOUR OWN CASE                          *
**********************************************************************
```

```
>>rename resnew res /home/ytli/tmp/*/*
```

We now run `rci_mpi` on four processors in restart mode, see ([27], Section 6.7). The restart mode means that the full Hamiltonian is read from the restart files and the only operation of the program is to diagonalize the Hamiltonian matrix.

```
***********************************************************************
*    RUN RCI_MPI IN RESTART MODE USING 4 PROCESSORS                   *
*    INPUT FILES: isodata, 2s2p.c, 2s2p.w                             *
*                  rci000.res, rci001.res, rci002.res, rci003.res     *
*    OUTPUT FILES: 2s2p.cm, 2s2p.csum, 2s2p.clog                      *
*    THIS IS A RESTART THAT READS THE rcixxx.res file                 *
***********************************************************************

>>mpirun -np 4 rci_mpi

 =======================================================
        RCI_MPI: Execution Begins ...
 =======================================================
 Participating nodes:
   Host: node84    ID: 000
   Host: node84    ID: 001
   Host: node84    ID: 002
   Host: node84    ID: 003

 Start Dir:
   node84: /home/ytli/data_node238/CIII/mpitest
   node84: /home/ytli/data_node238/CIII/mpitest
   node84: /home/ytli/data_node238/CIII/mpitest
   node84: /home/ytli/data_node238/CIII/mpitest

 Serial I/O Dir (node-0 only):
   node84: /home/ytli/data_node238/CIII/mpitest

 Work Dir (Parallel I/O):
   node84: /home/ytli/tmp
   node84: /home/ytli/tmp
   node84: /home/ytli/tmp
   node84: /home/ytli/tmp

 Default settings?
>>n
 Name of state:
>>2s2p
 Block            1 ,  ncf =         9744
 Block            2 ,  ncf =        27532
 Block            3 ,  ncf =        40692
 Loading CSF file ... Header only
 There are/is         49  relativistic subshells;
 Restarting RCI90 ?
>>y
 Estimate contributions from the self-energy?
>>n

.........
```

```
mpi stopped by node-          0  from RCI_MPI: Execution complete.
mpi stopped by node-          1  from RCI_MPI: Execution complete.
mpi stopped by node-          2  from RCI_MPI: Execution complete.
mpi stopped by node-          3  from RCI_MPI: Execution complete.
```

To display the energy separations after fine-tuning, we use the `rlevels` program, see ([27], Section 6.1).

```
*********************************************************************
*    READ THE *.cm FILES AND DISPLAY THE ENERGIES                  *
*********************************************************************

>>rlevels 2s2.cm 2s2p.cm

 nblock =       1  ncftot =      9788  nw =      49  nelec =      4
 nblock =       3  ncftot =     77968  nw =      49  nelec =      4

 Energy levels for ...
 Rydberg constant is   109737.31569
 Splitting is the energy difference with the lower neighbor
 --------------------------------------------------------------------

 No Pos  J Parity Energy Total    Levels      Splitting     Configuration
                    (a.u.)       (cm^-1)      (cm^-1)

 --------------------------------------------------------------------

   1   1  0  +     -36.5052083       0.00         0.00  1s(2).2s(2)_1S
   2   1  0  -     -36.2666030   52367.82     52367.82  1s(2).2s_2S.2p_3P
   3   1  1  -     -36.2664960   52391.30        23.48  1s(2).2s_2S.2p_3P
   4   1  2  -     -36.2662396   52447.57        56.28  1s(2).2s_2S.2p_3P
   5   2  1  -     -36.0388389  102356.24     49908.67  1s(2).2s_2S.2p_1P

 --------------------------------------------------------------------
```

The energies are now in very good agreement with the experimental energies from NIST. The remaining differences are due to interactions that have not been catered for. If deemed important, the fine-tuning can be redone to improve the energies further.

## 11. Applications

In this section, we apply fine-tuning for the computation of the $2s^2\,^1S_0 - 2s2p\,^{1,3}P_1$ transition rates in C III and for investigating the effect of the $2s2p^2\,^2S_{1/2}$ perturber on the lifetimes of the $2s^2ns\,^2S_{1/2}$ ($n = 3 - 7$) Rydberg states in B I.

### 11.1. Transition Rates in C III

The calculations for C III proceed as in the test-run. The radial orbitals were generated in EOL MCDHF calculations optimizing simultaneously on the even and odd state levels based on CSF expansions generated by SD excitations to an active set of orbitals with the restriction that the $1s^2$ core electrons are kept fixed. The active set of orbitals was increased layer-by-layer up to $n = 7$ ($\{7s, 7p, \ldots, 7h, 7i\}$), resulting in 133 CSFs of even parity and 760 CSFs of odd parity. The MCDHF calculations were followed by separate `rci_mpi` calculations for the even and odd state CSF expansions generated by SDT excitations to an active set of orbitals with the restriction that there should never be more than one excitation from the $1s^2$ core. The `rci_mpi` calculation includes the Breit interaction. The resulting expansions consisted of 9 788 and 77 968 CSFs for even and odd parity, respectively. Fine-tuning was applied to all `rci_mpi` calculations for the odd states based on the $2s2p$ MR. The excitation energies and the root-mean-square deviations, $\sigma_M$, between our results and the experimental energies from the NIST database [28] are shown in Table 2 as functions

of the increasing set of orbitals. The fine-tuned energies are all close to the experimental energies from the NIST database [28]. The ab initio energies, especially the energy for $2s2p\ ^1P_1$, are rather far away from the experimental energies for the small orbital sets ($n = 3, 4$), but improve as the orbital set is increased. Even for $n = 7$, some energy differences remain for the ab initio calculation. For variational calculations, the convergence of the energy differences is not always smooth, but the differences may increase or decrease a little between the orbital layers depending on where the orbitals are localized. In this case we see that the excitation energies become a little worse when going from $n = 6$ to $n = 7$. The overall trend is however to approach the experimental energies. The tendency for the energy differences from ab initio calculations to jump up and down is a further argument for fine-tuning.

**Table 2.** Convergence of C III excitation energies in (cm$^{-1}$) from ab initio and fine-tuned CI calculations based on increasing sets of orbitals. The root-mean-square deviations, $\sigma_M$, between our results and the experimental energies from the NIST database are also given.

| State | Ab Initio | | | | | Fine-Tuned | | | | | NIST |
|---|---|---|---|---|---|---|---|---|---|---|---|
| | $n \leq 3$ | $n \leq 4$ | $n \leq 5$ | $n \leq 6$ | $n \leq 7$ | $n \leq 3$ | $n \leq 4$ | $n \leq 5$ | $n \leq 6$ | $n \leq 7$ | |
| $2s^2\ ^1S_0$ | 0 | 0 | 0 | 0 | 0 | 0 | 0 | 0 | 0 | 0 | 0 |
| $2s2p\ ^3P_0$ | 52,321.81 | 52,445.12 | 52,411.16 | 52,407.97 | 52,463.43 | 52,366.94 | 52,367.32 | 52,367.24 | 52,367.23 | 52,367.82 | 52,367.06 |
| $2s2p\ ^3P_1$ | 52,344.14 | 52,468.08 | 52,434.34 | 52,431.32 | 52,486.91 | 52,390.57 | 52,391.08 | 52,390.92 | 52,390.89 | 52,391.30 | 52,390.75 |
| $2s2p\ ^3P_2$ | 52,398.05 | 52,523.24 | 52,490.00 | 52,487.35 | 52,543.19 | 52,446.97 | 52,447.43 | 52,447.27 | 52,447.31 | 52,447.57 | 52,447.11 |
| $2s2p\ ^1P_1$ | 104,112.00 | 102,962.02 | 102,613.81 | 102,517.46 | 102,530.94 | 102,395.09 | 102,366.93 | 102,358.33 | 102,356.05 | 102,356.24 | 102,352.04 |
| $\sigma_M$ | 880.92 | 312.23 | 136.21 | 89.86 | 122.24 | 21.53 | 7.45 | 3.15 | 2.01 | 2.16 | |

As discussed in Section 4, fine-tuning affects the calculated transition parameters in two ways: through the corrected transition energies $\Delta E$ that enter as multiplicative factors and through the modification of the expansion coefficients. The transition rates for $2s^2\ ^1S_0 - 2s2p\ ^1P_1$ and $2s^2\ ^1S_0 - 2s2p\ ^3P_1$ are shown in Table 3 as functions of the increasing set of orbitals. In accordance with what has been reported in the literature for MCHF-BP calculations, the convergence to final values are faster for the fine-tuned calculations compared with the ab initio calculations. In this case, we see no tendency of the fine-tuning process to "over-correct" for inaccuracies in energy separations, as is sometimes the case [1].

**Table 3.** The $2s^2\ ^1S_0 - 2s2p\ ^{1,3}P_1$ transition rates in C III as functions of the increasing orbital set from ab initio and fine-tuned CI calculations.

| Method | Orbitals | $2s^2\ ^1S_0 - 2s2p\ ^1P_1$ | $2s^2\ ^1S_0 - 2s2p\ ^3P_1$ |
|---|---|---|---|
| ab initio | $n \leq 3$ | $1.86 \times 10^9$ | $8.73 \times 10^1$ |
| | $n \leq 4$ | $1.80 \times 10^9$ | $9.59 \times 10^1$ |
| | $n \leq 5$ | $1.78 \times 10^9$ | $1.00 \times 10^2$ |
| | $n \leq 6$ | $1.77 \times 10^9$ | $1.02 \times 10^2$ |
| | $n \leq 7$ | $1.77 \times 10^9$ | $1.03 \times 10^2$ |
| fine-tuned | $n \leq 3$ | $1.77 \times 10^9$ | $9.56 \times 10^1$ |
| | $n \leq 4$ | $1.77 \times 10^9$ | $9.80 \times 10^1$ |
| | $n \leq 5$ | $1.77 \times 10^9$ | $1.01 \times 10^2$ |
| | $n \leq 6$ | $1.77 \times 10^9$ | $1.02 \times 10^2$ |
| | $n \leq 7$ | $1.77 \times 10^9$ | $1.03 \times 10^2$ |
| experiment | | $1.77 \times 10^9$ [a] | $1.0294(14) \times 10^2$ [b] |

[a] NIST [28], [b] Reference [29].

## 11.2. Lifetimes of the $2s^2ns\ ^2S_{1/2}$ Rydberg States in B I

The energy level structure for B I, given in Figure 1, is similar to that of a one-electron system. The ground configuration is $1s^22s^22p$, and higher states are formed by excitation of the $2p$ electron. In addition, the $1s^22s2p^2$ configuration gives rise to a few states below the ionization energy. The presence of these states, especially $1s^22s2p^2\ ^2S_{1/2}$, causes pertur-

bations of close-lying states of the same symmetry. This can be observed as irregularities in the energy positions of the levels. The perturbation can also have a strong effect on lifetime values. In an unperturbed sequence, lifetime values can be expected to increase regularly with the principal quantum number. There are more electron correlation for the states of the $1s^2 2s 2p^2$ perturber than there are for the Rydberg states, and very large and balanced calculations are required to place the perturbers at the right position. This is a case where fine-tuning is very valuable.

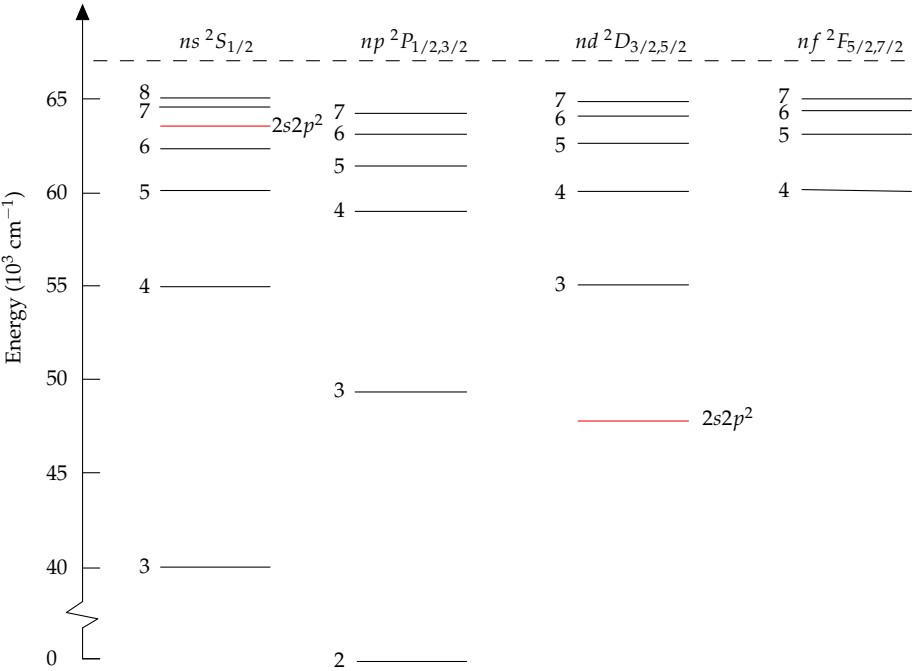

**Figure 1.** Energy-level diagram for neutral boron. The $2s 2p^2\ ^2S_{1/2}$ perturber is embedded in the $ns\ ^2S_{1/2}$ Rydberg series, with high impact on the lifetimes of the nearby states.

The radial functions of B I are from EOL MCHF calculations using the ATSP2K package [10]. Separate calculations were performed for the $^2S$ and $^2P$ states. The calculations for $^2S$ targeted the $2s^2 ns\ ^2S$ ($n = 3 - 7$) Rydberg states and the $2s 2p^2\ ^2S$ perturbing state and were based on complete active space (CAS) expansions with $1s^2$ as a closed core. The orbital set was systematically increased to $\{12s, 8p, 8d, 8f, 8g, 7h\}$. The calculations for $^2P$ targeted the $2s^2 np\ ^2P$ ($n = 3 - 6$) Rydberg states and were based on CAS expansions with $1s^2$ as a closed core. The orbital set was systematically increased to $\{8s, 12p, 8d, 8f, 8g, 7h\}$. The `rwfnmchfmcdf` program was used to convert the non-relativistic radial orbitals to relativistic ones in GRASP format, see [27], Sections 3.3 and 6.2. The MCHF calculations were followed by separate `rci_mpi` calculations for the even and odd state CSF expansions, accounting also for the Breit interaction. The energies from the `rci_mpi` calculations are displayed in Table 4 from which we see that the $2s 2p^2\ ^2S_{1/2}$ perturbing state is too high and in the wrong position in the Rydberg series. To correct for this, fine-tuning was applied for the even states based on the $2s^2 ns\ ^2S_{1/2}$ ($n = 3 - 7$) and $2s 2p^2\ ^2S_{1/2}$ MR. Even in $LSJ$-coupling, the states are heavily mixed with relatively large off-diagonal matrix elements, see [8]. Due to this strong mixing, the fine-tuning had to be carried out iteratively before a good agreement with the experimental energies were found. The fine-tuned energies are displayed in Table 4. Now the agreement with the experimental energies from the NIST database [28] is good, and the perturber is in the correct position between the $6s\ ^2S_{1/2}$ and $7s\ ^2S_{1/2}$ Rydberg states.

The position of the perturber has a dramatic effect on the computed lifetimes of the $^2S$ states as is seen in Table 5. In the ab initio calculation, the $2s 2p^2\ ^2S_{1/2}$ short-lived perturber is too high relative to the $2s^2 ns\ ^2S_{1/2}$ ($n = 3 - 7$) Rydberg states, which results

in erroneously weak mixing into the latter. The weak mixing leads to lifetimes of the Rydberg states that are too long. The fine-tuning brings the perturber into the correct position with a corresponding increased mixing into the $2s^2ns\,^2S_{1/2}$ ($n = 3 - 7$) Rydberg states, which are now considerably more short-lived, in good agreement with the lifetime measurements by Lundberg et al. [8] using laser-induced fluorescence. The effect of the increased mixing is more pronounced for the higher Rydberg states, but can be traced back also to the low $2s^23s\,^2S_{1/2}$ and $2s^24s\,^2S_{1/2}$ states. The effect of the fine-tuning on the lifetime of the perturber itself is small.

**Table 4.** Energies (in cm$^{-1}$) for B I from ab initio and fine-tuned CI calculations compared with experimental energies from the NIST database [28].

| State | Ab Initio | Fine-Tuned | NIST |
|---|---|---|---|
| $1s^22s^22p\,^2P^o_{1/2}$ | 0 | 0 | 0 |
| $1s^22s^22p\,^2P^o_{3/2}$ | 14.51 | | 15.287 |
| $1s^22s^23s\,^2S_{1/2}$ | 39,800.91 | 40,038.65 | 40,039.6907 |
| $1s^22s^23p\,^2P^o_{1/2}$ | 48,520.93 | | 48,611.8663 |
| $1s^22s^23p\,^2P^o_{3/2}$ | 48,522.60 | | 48,613.6486 |
| $1s^22s^24s\,^2S_{1/2}$ | 54,707.61 | 55,012.88 | 55,010.2338 |
| $1s^22s^24p\,^2P^o_{1/2}$ | 57,669.26 | | 57,786.4336 |
| $1s^22s^24p\,^2P^o_{3/2}$ | 57,669.86 | | 57,787.0683 |
| $1s^22s^25s\,^2S_{1/2}$ | 59,860.42 | 60,146.56 | 60,146.414 |
| $1s^22s^25p\,^2P^o_{1/2}$ | 61,306.69 | | 61,433.59 |
| $1s^22s^25p\,^2P^o_{3/2}$ | 61,306.97 | | 61,433.59 |
| $1s^22s^26s\,^2S_{1/2}$ | 62,254.45 | 62,476.47 | 62,482.167 |
| $1s^22s^26p\,^2P^o_{1/2}$ | 63,129.71 | | 63,263.24 |
| $1s^22s^26p\,^2P^o_{3/2}$ | 63,129.87 | | 63,263.24 |
| $1s^22s2p^2\,^2S_{1/2}$ | 65,102.64 | 63,464.94 | 63,560.638 |
| $1s^22s^27s\,^2S_{1/2}$ | 63,526.70 | 64,166.09 | 64,156.017 |

**Table 5.** Lifetimes (in ns) for the $2s^2ns\,^2S_{1/2}$ Rydberg sequence in neutral boron from ab initio and fine-tuned CI calculations compared with experimental lifetimes and lifetimes from other theory.

| State | Ab Initio | Fine-Tuned | Experiment | MCHF |
|---|---|---|---|---|
| $2s^23s\,^2S_{1/2}$ | 4.11 | 3.99 | 4.0 (2) [a] | 3.97 [b] |
| $2s^24s\,^2S_{1/2}$ | 9.86 | 8.67 | 8.7 (4) [a] | 8.59 [b] |
| $2s^25s\,^2S_{1/2}$ | 17.6 | 12.2 | 11.0 (6) [b] | 11.3 [b] |
| $2s^26s\,^2S_{1/2}$ | 25.1 | 8.40 | 7.7 (4) [b] | 7.65 [b] |
| $2s2p^2\,^2S_{1/2}$ | 3.24 | 3.29 | 3.3 (2) [b] | 3.65 [b] |
| | | | 3.6 (3) [c] | |
| $2s^27s\,^2S_{1/2}$ | 23.6 | 11.1 | 8.3 (4) [b] | 8.01 [b] |

[a] Reference [30], laser-induced fluorescence. [b] Reference [8]; experiment selective laser excitation and theory fine-tuned MCHF calculations. [c] Reference [31], beam-foil.

## 12. Heavy and Complex Systems

The applications in Section 11 were for two light systems C III and B I, mainly because highly accurate experimental values of transition rates and lifetimes are available for comparison. The real benefit of fine-tuning is however for more heavy and complex systems with dense energy structures, and the new programs `jj2lsj_2022` and `rfinetune` are applicable also in these cases. As for the CPU time for fine-tuning, this mainly depends on the number of CSFs and the number of tuning iterations needed to reach a good agreement with the experimental energies. Assuming a case with around 2.2 million CSFs for a single *J*-block, the construction of the Hamiltonian matrix takes roughly 1400 s using 16 processors on an AMD EPYC 7763 Linux server, see Table 6 in [32]. The diagonalization takes an additional 200 s making a total of around 1600 s for the initial `rci_mpi` run. Even for spectrum calculations for heavy and complex systems, the number of CSFs in the MR is small and the transformation from *jj*- to *LSJ* using `jj2lsj_2022` is performed in seconds on a single processor. The subsequent execution of `rfinetune` to produce the transformed Hamiltonian submatrix that is merged into the restart files of the initial

`rci_mpi` calculation also finishes within seconds on a single processor. The only non-negligible time for fine-tuning is the additional time for running `rci_mpi` in restart mode. This amounts to diagonalizing the modified Hamiltonian matrix, which in this case takes around 200 s. If four iterations are needed to reach a good agreement with the experimental energies, the additional time for fine-tuning is around 800 s. This time corresponds to 50% of the time for the initial `rci_mpi` calculation.

## 13. Summary and Conclusions

Fine-tuning of diagonal elements of the Hamiltonian matrix is an efficient way to ensure that calculated energy separations of states which mix are in agreement with experiment. The process gives more accurate measures of the mixing than can be obtained in ab initio calculations and results in improved convergence of computed properties such as transition rates as the calculation is enlarged. Fine-tuning assumes that off-diagonal matrix elements are small compared with the differences between diagonal elements. This assumption is often not fulfilled for relativistic calculations in *jj*-coupling. We show that these problems can be overcome with a transformation of the Hamiltonian submatrix corresponding to the CSFs in the MR from *jj*- to *LSJ*-coupling. The method is implemented in the GRASP2018 package [17] through the new programs `jj2lsj_2022` and `rfinetune`. Transformation from *jj*- to *LSJ*-coupling is not the only option, but an improved methodology would rely on a transformation from *jj*-coupling to the *optimal* coupling (*LSJ*, *JK*, *LK* etc.), in which the states are as pure as possible. Work along these lines are in progress based on the `Coupling` program by Gaigalas [33]. Even in optimal coupling, some states may be considerably mixed due to close degeneracies, and fine-tuning then becomes a non-linear problem that needs to be solved iteratively. The best approach in this case would be to see fine-tuning as a non-linear least squares problem that can be solved by interfacing the current programs with a secant-based least-squares code such as `NL2SOL` [34]. A final question relates to which states to fine-tune and by how much. To answer this question we compute the mean level deviation (*MLD*)

$$MLD = \frac{1}{N} \sum_{i=1}^{N} |E_{\exp}(i) - E_{\text{calc}}(i) + ES|, \tag{49}$$

where the energy shift (*ES*) is chosen as to minimize the sum. *ES* indicates to what extent the ground state level is relatively too low (*ES* negative) or too high (*ES* positive) in the theoretical binding energy balance [35]. Once *ES* is determined, the ground state may be fine-tuned by $-ES$ and the remaining states by $E_{\exp}(i) - E_{\text{calc}}(i) + ES$.

**Author Contributions:** Methodology, Y.L., G.G., W.L., C.C. and P.J.; software, Y.L., G.G. and P.J.; validation, Y.L., G.G., W.L., C.C. and P.J.; investigation, Y.L., G.G., W.L., C.C. and P.J.; writing—original draft, Y.L. and P.J.; writing—review and editing, Y.L., G.G., W.L., C.C. and P.J.; visualization, P.J. All authors have read and agreed to the published version of the manuscript.

**Funding:** Y.L. and C.C. acknowledge support from the National Natural Science Foundation of China (Grant nos. 12104095 and 12074081). Y.L. acknowledges support from the China Scholarship Council with Grant No. 202006100114.

**Data Availability Statement:** Not applicable.

**Acknowledgments:** The authors dedicate the paper to Alan Hibbert, whose pioneering work on fine-tuning has been a source of inspiration and something to strive for.

**Conflicts of Interest:** The authors declare no conflict of interest.

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
