# Peer review of "Fine-Tuning of Atomic Energies in Relativistic Multiconfiguration Calculations"

_atoms, doi:10.3390/atoms11040070_

Round 1

Reviewer 1 Report

In this paper, the authors presents a procedure for obtaining more accurate measures of atomic energies and computed properties compared to ab initio methods. This fine-tuning process involves applying a correction to the diagonal elements of the Hamiltonian, assuming the Hamiltonian is diagonally dominant. The implementation of this method is actualized in the GRASP2018 code package and has been applied to calculations involving transition rates in C III and lifetimes of Rydberg transitions in B I. The paper is well written and would be a great addition to the special issue of GRASP in Atoms. The authors introduce the theory and methods in great detail with references, as well as concrete examples of the methods including code execution.

Several comments/possible revisions follow:

- The methods described here has been applied to light systems with simple atomic structure. Has there been any tests or applications to heavier systems? Curious how the method would scale with larger matrix sizes. 

- It would be interesting to have a paragraph about the computation times required for some of these calculations with fine-tuning. For example, in the case of fine-tuning in jj-coupling where there are large off-diagonal matrix elements, the procedure requires two complete relativistic CI calculations to be performed. How does the runtime of the fine-tuning procedure typically compare with ab initio calculations? Or are the systems tested here not large enough for these considerations to be taken into account?

- Would the fine-tuned calculations require the same number of executions to ensure convergence as the ab initio calculation? Referencing Table 3, calculations were done with increasing orbitals sets up to n <= 7 for both cases. While the fine-tuned transition rates seemed to converge as early as the n <=3, n <= 4 calculations, for the 1S0 - 3P1 transition, they both took up to the n <= 7 calculation to find the closest agreement with experiment. How would this be done in practice? 

Some minor grammar errors:

- line 518: ab inition -> ab initio

- line 538: ... perturber than there is ... -> ... perturber than there are ...

Reviewer 2 Report

Q1:  Several paper authors claim that they made some improvements based on GRASP2018 or GRASP2k. But their program was not published for free or for open. In this paper, the new code was also introduced, How long it can be made for open?

Q2Using JJ2LSJ code, a problem was encountered, For example, there were 56 CSFs of 3s3p^23d. Few CSFs were missing in the composition for some cases. How to solve it?

Q3: How many time and memory needed in this run?

Round 2

Reviewer 1 Report

Thank you for the revisions, no further comments.

Reviewer 2 Report

Although these authors are well known, they also answer questions carefully. I think this paper can be published.